# The EGFR-TMEM167A-p53 Axis Defines the Aggressiveness of Gliomas

**DOI:** 10.3390/cancers12010208

**Published:** 2020-01-14

**Authors:** Berta Segura-Collar, Ricardo Gargini, Elena Tovar-Ambel, Esther Hernández-SanMiguel, Carolina Epifano, Ignacio Pérez de Castro, Aurelio Hernández-Laín, Sergio Casas-Tintó, Pilar Sánchez-Gómez

**Affiliations:** 1Neurooncology Unit, Instituto de Salud Carlos III-UFIEC, Crtra/Majadahonda-Pozuelo, Km 2, Majadahonda, 28220 Madrid, Spain; berta.segura@externos.isciii.es (B.S.-C.); rgargini@isciii.es (R.G.); etovarambel@gmail.com (E.T.-A.); estherhsm@hotmail.com (E.H.-S.); 2Centro de Biología Molecular “Severo Ochoa” (CSIC-UAM), 28220 Madrid, Spain; 3Fundación Andrés Marcio “Niños contra la laminopatía”, 28220 Madrid, Spain; cepifano@hotmail.com; 4Gene Therapy Unit, Instituto de Salud Carlos III-IIER, 28220 Madrid, Spain; iperez@isciii.es; 5Instituto de investigaciones Biomédicas I + 12, Hospital 12 de Octubre, 28220 Madrid, Spain; aurelio.hlain@salud.madrid.org; 6Instituto Cajal, Centro Superior de Investigaciones Científicas (CSIC), 28220 Madrid, Spain; scasas@cajal.csic.es

**Keywords:** glioma, wild-type p53, mutant p53, EGFR/AKT signaling, vesicular trafficking, autophagy

## Abstract

Despite the high frequency of *EGFR* and *TP53* genetic alterations in gliomas, little is known about their crosstalk during tumor progression. Here, we described a mutually exclusive distribution between mutations in these two genes. We found that wild-type p53 gliomas are more aggressive than their mutant counterparts, probably because the former accumulate amplifications and/or mutations in *EGFR* and show a stronger activation of this receptor. In addition, we identified a series of genes associated with vesicular trafficking of EGFR in p53 wild-type gliomas. Among these genes, *TMEM167A* showed the strongest implication in overall survival in this group of tumors. In agreement with this observation, inhibition of *TMEM167A* expression impaired the subcutaneous and the intracranial growth of wild-type p53 gliomas, regardless of the presence of *EGFR* mutations. In the absence of p53 mutations, *TMEM167A* knockdown reduced the acidification of intracellular vesicles, affecting the autophagy process and impairing EGFR trafficking and signaling. This effect was mimicked by an inhibitor of the vacuolar ATPase. We propose that the increased aggressiveness of wild-type p53 gliomas might be due to the increase in growth factor signaling activity, which depends on the regulation of vesicular trafficking by TMEM167A.

## 1. Introduction

Gliomas are the most frequent primary tumors of the brain. Patients with grade IV gliomas (glioblastomas, GBMs) have a very unfavorable prognosis, with a mean survival of 16.6 months. This is due in large part to limitations in surgical resection and to the lack of effective treatments that could avoid the inexorable recurrence of these lethal tumors. 

The comprehensive analysis of genomic and epigenomic deregulations in gliomas has shown that it is a heterogeneous disease with distinct tumor subtypes. Mutations in the *Isocitrate dehydrogenase 1/2* (*IDH1/2*) genes, for example, generate gliomas with a better clinical prognosis. By contrast, *Epidermal growth factor receptor* (*EGFR*) alterations have been implicated in the malignant progression of this type of cancer. *EGFR* is amplified and/or mutated in nearly 50% of high-grade gliomas, favoring the survival of tumor cells and increasing their proliferative, angiogenic, and invasive capacities [1,2]. Other common mutations occur in the p53 pathway, in the Rb regulatory circuit, as well as in other Tyrosine-kinase receptors or in their downstream effectors [3]. However, even though the research has focused on the characterization of these different genetic alterations, little is known about the interaction between them. 

Mutations in *TP53* are a hallmark of cancer. These mutations abrogate the tumor-suppressor functions of wild-type p53, but they also endow the mutant protein with novel oncogenic activities. One of the gain-of-functions (GOFs) of mutant p53 is related to its capacity to sequester the transcription factor p63, which indirectly results in enhanced recycling of several receptors back to the plasma membrane, boosting the activation of downstream signals such as AKT. This function has been well established for several p53 mutants and different membrane receptors, including EGFR, and has even been demonstrated in GBM [4,5]. Among other activities, the presence of stabilized mutant p53 isoforms result in metabolic changes through interaction with Sterol regulatory element-binding proteins (SREBP), promote interaction with DNA damage regulators on the Ataxia telangiectasia mutated (ATM) pathway, or generate resistance to DNA damage by associating with the Nuclear factor Y (NF-Y) [4]. The detection of *TP53* mutations is associated with a worse prognosis for different tumor types. In fact, several therapeutic strategies are being developed to target the stability or the function of the mutant proteins [6]. Conversely, mutations in *TP53* has been attributed to the initial steps of gliomagenesis, where they tend to co-occur with mutations in *IDH1/2*, which accumulate in lower-grade gliomas (LGG) and the less aggressive GBMs [3,7].

Consistent with the oncogenic role of EGFR in GBM, the main genetic alterations observed in these tumors are chromosomal amplifications and point mutations, which lead to hyperactivation of the receptor. The latter ones include *n*-terminal deletions that relieves the EGFR dependency on extracellular ligands. Additional mechanisms contribute to increase EGFR signaling, like interaction with other receptors and amplification or overexpression of extracellular ligands [1,2]. Moreover, several endocytic/recycling molecules contribute to the stabilization of EGFR in gliomas and other cancers, increasing the robustness of the signaling cascade and/or relocating the activation of the downstream targets [8,9]. 

Here, we set out to study the possible association between the genetic status of *TP53* and *EGFR* in gliomas. Moreover, we investigated if mutant p53 proteins could participate in the stabilization of the receptor, which is essential for the progression of this type of tumors. Our data confirmed that gliomas expressing mutant p53 are less aggressive than the wild-type p53 tumors. Moreover, we showed that the latter accumulate alterations in *EGFR* and have a greater receptor activity. Based on that, we searched for other molecules that could modulate the trafficking of membrane receptors in wild-type p53 GBMs. We focused on *TMEM167A*, as we had previously linked this gene to EGFR regulation in gliomas [10]. We measured a strong association between the expression of this gene and the worse survival of patients with wild-type p53 tumors. Moreover, we found that *TMEM167A* inhibition in this subgroup of GBMs reduced the EGFR–AKT signaling axis and impaired tumor growth. This effect was rescued in the presence of mutant p53 proteins. *TMEM167A* knockdown (KD) reduced acidification of the endo-lysosomal vesicles, which blocked EGFR-induced AKT activation and inhibited the autophagy processes. Moreover, treatment with bafilomycin A1 (BFA), a vacuolar ATPase inhibitor that impairs vesicular acidification, reproduced the effects of *TMEM167A* downregulation in the EGFR/AKT signaling. Together, our data suggest that the increased aggressiveness of wild-type p53 gliomas is associated with higher EGFR/AKT activity, which depends on the regulation of vesicular acidification and function exerted by TMEM167A. 

## 2. Results

### 2.1. p53 Wild-Type Gliomas Are More Aggressive and Have Stronger EGFR Signaling

To study the relation between the genetic status of TP53 and EGFR in gliomas we performed an in silico analysis, using the TCGA cohort. We first confirmed that patients with wild-type p53 gliomas have a decreased overall survival in comparison with patients with mutant p53 tumors (Figure 1A). Moreover, we found a positive correlation between TP53 mutations and those occurring in IDH1 and ATRX genes (Figure 1B), which tend to accumulate in LGG. Considering that p53 mutant proteins regulate EGFR recycling, we proposed a positive relation between the alterations in both genes. However, the data showed that there is a mutually exclusive distribution between mutations in TP53 and mutations or amplifications in EGFR (Figure 1B–D). Moreover, we observed an increase in receptor phosphorylation p53 wild-type gliomas (Figure 1E). These results were replicated when we analyzed the GBM-only cohort, where TP53 mutations were also associated with a better prognosis (Appendix A) and inversely correlated with mutations (Appendix A) and amplifications (Appendix A) in EGFR. Furthermore, we found an accumulation of phosphorylated EGFR in p53 wild-type versus. mutant GBM samples analyzed by Western blot (WB) (Figure 1F). Together, these data indicate that activation of EGFR signaling occurs mostly in p53 wild-type tumors, which could contribute to their increased aggressiveness.

### 2.2. Analysis of Vesicular Transport-Related Genes Associated with EGFR in Wild-Type p53 Gliomas

Regulation of EGFR trafficking is a key factor in the oncogenic functions of this receptor. Molecules that promote the recycling or inhibit the degradation, increase the activation of EGFR and its downstream readouts [8]. However, the mutually exclusive relation observed between mutations in TP53 and EGFR in gliomas suggests that other mechanisms, different from GOF activities of mutant p53 proteins, could be responsible for the regulation of EGFR turnover in the most aggressive tumors, those that accumulate alterations in the receptor’s gene. As an alternative, we considered DYRK1A (dual-specificity tyrosine phosphorylation-regulated kinase 1A), as we have previously demonstrated that this kinase promotes the stability of EGFR in adult neural progenitors [11] and glioma cells, where it controls the growth and the survival of the tumors [12]. However, the DYRK1A expression decreased with tumor grade (Appendix A). Moreover, it correlated positively with an increase in the overall survival of patients with glioma (Appendix A), so it does not seem to have a preponderant role in the most aggressive tumors. Therefore, we performed an unbiased screening to find molecules involved in vesicular transport in wild-type p53 gliomas, which showed higher levels of EGFR transcription (Appendix A). For that, we separated the TCGA glioma cohort in two groups (with low- and high-quantity of EGFR mRNA), and we analyzed the differential expression of a signature of genes associated with endosomes and vesicular transport (Appendix A). We chose the three genes most significantly enriched in the high-EGFR gliomas: GOSR1, TMEM167A, and STX17 (Appendix A). We then confirmed that the three of them were overexpressed in the gliomas with respect to normal tissue (Figure 2A–C). However, among the three vesicular-related genes, the expression of TMEM167A showed the highest correlation with the survival of the patients (Figure 2D–F) (average survival of High and Low TMEM167A subgroups was 396 versus. 837 days, respectively). These results were confirmed in our own cohort of 52 glioma samples (Figure 2G,H), suggesting that this gene might regulate glioma aggressiveness. TMEM167A is a transmembrane protein that has been associated with the Golgi apparatus and with the vesicle secretion processes [13]. Recently, we have shown that the downregulation of TMEM167A or its fly orthologue, Kish, diminish glioma growth by affecting the endo-lysosomal system. Moreover, our previous data suggested that the downregulation of TMEM167A/Kish induces degradation of EGFR [10]. However, we had not explored the relation between TMEM167A expression and/or function with the genetic status of EGFR and TP53.

### 2.3. TMEM167A Controls the Growth of Wild-Type p53 GBM but It Is Dispensable for Mutant p53 Tumors

To begin to elucidate the relation of TMEM167A function with the genetic status of *TP53*, we performed an in silico analysis. The association of *TMEM167A* transcription with a poor prognosis was only significant in wild-type p53 gliomas (Figure 3A,B). Moreover, TMEM167A expression was lower in mutant compared with wild-type p53 tumors (Appendix A) and xenografts (Appendix A). All these results suggest that the pro-oncogenic function of this gene is no longer needed in mutant p53 tumors. To explore this hypothesis, we inhibited TMEM167A expression in different GBM lines (carrying wt or mutant p53) implanted in the flanks of immunodeficient mice (Appendix A). The expression of two different shRNA sequences in U87 (p53wt) cells (Appendix A) inhibited tumor growth (Figure 3C). However, TMEM167A KD (using the most effective shRNA sequence) in U373 (p53 mut) cells, as well as in two primary cell lines with mutations in p53, did not inhibit the growth of these tumor cells (Figure 3D–F).

We analyzed the effect of TMEM167A KD in the different GBM cells, and we observed that it only reduced amount of membrane EGFR in U87 cells (p53 wt) (Figure 4A), whereas there were no changes in the mutant p53 cell lines (Figure 4B–D). EGFR reduction was associated with the loss of receptor signaling (measured as the amount of AKT phosphorylation) in U87 cells (Figure 4E), which was not observed in the different p53mut cells (Figure 4F–H). These results indicate that EGFR stability and signaling depend on TMEM167A in wild-type p53 wt. However, in mutant p53 tumors, TMEM167A is dispensable for EGFR activity. 

### 2.4. TMEM167A Controls the Orthotopic Growth of Wild-Type p53 Gliomas, Independently of the Presence of Additional Mutations in EGFR

To get further insight into the function of TMEM167A in wild-type p53 GBMs, we performed intracranial injections of U87 cells expressing shControl (Control) or shTMEM167Aa (shTMEM167Aa) in the brain of nude mice (Appendix A). We observed an increase in the survival of the mice that had been injected with shTMEM167A cells in comparison with the control-injected mice (Figure 5A). Moreover, TMEM167A knock-down (KD) induced a decrease in the amount of Phospho-AKT (Figure 5B), confirming our previous results [10] and demonstrating an efficient inhibition of EGFR signaling after the in vivo downregulation of TMEM167A. Even though this established cell line overexpresses EGFR, U87 cells do not carry alterations in this gene. To obtain an independent confirmation of the role of TMEM167A in wild-type p53 glioma cells, we inhibited its transcription in an EGFR amplified primary cell line (GBM4). Moreover, GBM4 cells express the N-terminal deleted vIII isoform (the most common EGFR mutation in GBM). The KD of TMEM167A in these cells (Appendix A) also increased the survival of the injected mice (Figure 5C) and reduced AKT activation (Figure 5D). These results further support the relevance of TMEM167A function to maintain EGFR signaling in wild-type p53 gliomas and suggest that the antitumor effect of *TMEM167A* downregulation is independent on the presence of amplifications and/or mutations in the receptor’s gene. 

### 2.5. Expression of Mutant p53 Can Rescue the Effect of TMEM167A Downregulation on EGFR Signaling

To confirm the relation between TMEM167A function and p53, we performed a time course of stimulation with EGF and analyzed the activation of AKT in U87 (p53 wt) or U373 (p53 mut) cells in the presence of shControl or shTMEM167A (Appendix A). As expected, EGF induced a strong phosphorylation of AKT ten min after ligand incubation, which was restored to normal levels two hours later (Figure 6A). The response was similar in U87 and U373 cells. However, we observed that TMEM167A KD strongly impaired AKT phosphorylation in U87 cells, whereas there was no change in the response to EGF in the mutant p53 U373 cells (Figure 6A). Moreover, the overexpression of p53 R273H, a common mutant isoform, in U87 cells (Appendix A) rescued the Phospho-AKT reduction induced by shTMEM167A (Figure 6B and Appendix A), suggesting that, in the presence of a GOF of mutant p53, TMEM167A is no longer required for the EGFR/AKT signaling.

### 2.6. TMEM167A Is Required for Vesicular Acidification and EGFR/AKT Signaling 

We have recently shown that Kish, the Drosophila orthologue of TMEM167A, regulates the endosomal trafficking of EGFR. Downregulation of this fly gene impairs the acidification of the lysosomes, favoring EGFR degradation by the alternative proteasomal pathway [10]. To decipher the mechanism of action of TMEM167A in human cells, we first analyzed the subcellular location of this protein. As there are no antibodies available for this protein, we overexpressed a GFP-fused version of TMEM167A in 293T cells. We observed a strong colocalization of TMEM167A with acidic (LysoTracker positive) vesicles (Figure 7A). Moreover, the GFP-labeled protein colocalized with phosphatidylinositol-3,4-bisphosphate (PI(3,4)P2) molecules (Figure 7B). These phosphoinositides are accumulated on the plasma membrane and various endomembrane compartments, in response to the activation of several Tyrosine-kinase receptors. They act as docking sites for AKT recruitment and phosphorylation [14]. The presence of TMEM167A in the PI(3,4)P2-enriched vesicles suggests that this protein could be exerting a spatial regulation of AKT activation, downstream of EGFR signaling.

In a similar way to what occurs in fly cells, the KD of TMEM167A in U87 cells (p53 wt) cells produced a strong reduction of the number of acidic vesicles (Figure 7C). However, the LysoTracker staining did not change after TMEM167A downregulation in U373 cells (p53 mut) (Figure 7D). On the basis of these results, we wondered if TMEM167A inhibition could be impairing the autophagy process, which is totally dependent on the pH of the intracellular vesicles [15]. WB analysis of the cells treated with EGF in vitro showed a strong increase in the levels of LC3II and p62/SQSTM1 after TMEM167 downregulation in U87, but not in U373 cells (Appendix A). LC3II and p62 are two proteins that become degraded by the autophagosomes [15]. The accumulation of both markers after TMEM167A KD suggests that there is an inhibition of the degradative phase of the autophagy, which only occurs in p53 wild-type, but not in p53 mutant glioma cells. Moreover, we observed the accumulation of active cathepsin and p62 in U87 shTMEM167A compared with U87 control tumors (Appendix A), which is consistent with the increase in the pH and the decrease in the lysosomal-mediated protein degradation processes. A similar p62 accumulation (parallel to a reduction of Phospho-AKT) was measured in the intracranial U87 (Figure 7E) and GBM4 (Figure 7F) tumors after TMEM167A KD. This process has been widely described on autophagy-related studies, where cathepsin inhibitors, such as E64 and pepstatin, can block protein degradation, an effect that has been also linked to the lack of vesicle acidification [15]. In agreement with this, we were able to replicate the reduction of AKT phosphorylation and the accumulation of p62 by treating U87 cells with the vacuolar ATPase (v-ATPase) inhibitor, BFA (Figure 7G). In summary, our results demonstrate that TMEM167A is essential for the function of acidic vesicles, which are necessary for the maintenance of the EGFR–AKT axis in glioma cells. However, in the presence of p53 mutations acidification of the vesicles and, therefore, EGFR signaling are independent of TMEM167A. 

## 3. Discussion

Detection of p53 mutations is associated with a worse prognosis in most cancers [16]. By contrast, *TP53* mutations are early events in the development of gliomagenesis [17], and they accumulate in LGGs [7]. However, it is not clear why tumors expressing mutant p53 proteins have a less aggressive behavior, because most of the GOF attributed to these mutations generate advantages to tumor cells: cell cycle progression, metabolic adaptation, or even increased migration [18]. Here, we have re-evaluated how the genetic alterations in *TP53* influence glioma growth and progression. We observed that alterations in *EGFR* have a mutually exclusive distribution with mutations in *TP53*. It is interesting to note that such negative correlation is not observed in other tumors. In lung cancer patients, for example, tumors with *EGFR* mutations have a higher rate of mutant *TP53*. Moreover, the presence of these p53 mutations is a negative prognostic factor [19] and is even associated with a lower response to EGFR inhibitors in lung tumors [20]. By contrast, the results presented here, using the in silico data and our own cohort of samples, showed that there is a higher activation of EGFR signaling in wild-type p53 gliomas. This highlights the important differences between the mechanisms of EGFR function in gliomas versus. lung cancer [21,22]. Moreover, the results allow us to hypothesize that the activation of the receptor in the wild-type p53 gliomas could be responsible of the higher aggressiveness of these tumors. 

One of the oncogenic activities of mutant p53 is associated with increased recycling of membrane receptors [4,5]. Given the absence of such mutations in *EGFR* amplified and/or mutated tumors, we searched for other molecules that could favor EGFR stability and signaling in p53 wild-type gliomas. We identified a series of vesicular transport genes that are upregulated in gliomas with alterations in *EGFR*, which could have a direct implication in the functionality of this receptor. Among these genes, we picked *TMEM167A* due to its higher prognostic value. Moreover, a recent screening performed in fly glioma models had allowed us to identify Kish, the *Drosophila TMEM167A* orthologue, as a key modulator of EGFR trafficking/recycling during glioma development. Moreover, we had observed that the inhibition of *TMEM167A* expression affects the distribution of EGFR on the endo-lysosomal system [10]. Now, we have established that the prognostic value of *TMEM167A* expression is not relevant in mutant p53 gliomas. In agreement with this, EGFR/AKT signaling was dependent on *TMEM167A*, but only in wild-type p53 gliomas. AKT can be phosphorylated in response to different stimuli, especially after the activation of receptors with Tyrosine-kinase activity (RTKs) (like EGFR) or downstream of G-protein coupled receptors [23]. Although we have previously observed that the growth of most of the lines used in this study is EGFR-dependent [12,24], we cannot discard that TMEM167A might be necessary for the trafficking and/or the signaling of other receptors present in gliomas. It is also worth mentioning that the presence of mutant p53 made these gliomas independent of the EGFR–TMEM167A–AKT axis. We speculate that the alterations in the receptor’s turnover induced by mutations in p53 [4,5,6] might modify the mechanism of EGFR/AKT activation, making glioma cells insensitive to changes in the expression and/or the function of TMEM167. Other relevant mutations for glioma development involve the loss of PTEN (phosphatase and tensin homologue deleted from chromosome 10) expression or function [3], which activates the AKT pathway and, in particular, the production of PI (3,4)P2 [25]. However, our results indicate that the function of TMEM167A is independent of the *PTEN* status, as this gene is lost in both responsive (U87) and nonresponsive (U373) glioma cells. We could speculate that TMEM167A might function downstream of PTEN in the phosphoinositides-related signaling. 

TMEM167A is part of an extensive family of proteins that are anchored in the membrane. Several members of the TMEM family have been implicated in different cancer-related processes [26]. TMEM9, for example, has been established as a promoter of intestinal tumorigenesis by direct regulation of the v-ATPase, which favors the activation of the Wnt/-catenin pathway. The authors have described that TMEM9 binds to and facilitates assembly of the v-ATPase, being key for the maintenance of the pH in the acidic vesicles [27]. Similar to these results, the downregulation of *TMEM167A* strongly increased the pH of acidic vesicles. Moreover, BFA, which inhibits the v-ATPase function, was able to mimic the effect of *TMEME167A* KD on AKT activation. Together, these results suggest that several TMEM proteins might play an oncogenic function by maintaining the proper pH of the endomembrane system and, as a consequence, favoring vesicle-mediated signaling of different receptors. Moreover, their function could be relevant to maintain the high content of acidic vesicles found in gliomas, compared with normal glial cells [10]. Indeed, the presence of certain v-ATPase subunits has been shown to be enriched in tumor samples, where it correlated with shorter patients’ overall survival. Moreover, these authors described that BFA treatment or interference of the v-ATPase expression in GBM primary cell lines was able to inhibit the expression of stem cell markers [28]. It is interesting to note that the endosomal effect of *TMEM167A* KD was rescued in the presence of mutant p53 proteins. Although there is no GOF of p53 mutations directly linked to vesicular acidification, it is well established that there is an interplay between these mutant proteins and the autophagy and lysosomal-mediated degradation machinery [29]. This might also explain why *TMEM167A* KD did not change the Lysotracker staining and, therefore, EGFR signaling, in the presence of p53 mutations. 

Deregulation of the activity of different Tyrosine-kinase receptors is of paramount importance for cancer development. Although the internalization of these receptors upon ligand binding can target them for degradation in the lysosomes, it also serves to prolong the signaling or to modulate the specific activation of different targets in different locations. In fact, deregulation of endocytosis and vesicular trafficking has oncogenic potential in different cells [8]. One molecule that is essential for the degradation of EGFR and other receptors is RAB7, which is responsible for sorting them into late endosomes and lysosomes. In agreement with the effect observed after *TMEM167A* downregulation in gliomas cells, it has been shown that RAB7 is essential for AKT activation and signaling in other cancer cells [30]. Moreover, there is plenty of evidences supporting the existence of a tight regulation of the spatial and temporal compartmentalization of AKT activation and function [9]. 

The most frequent EGFR mutation is the EGFRvIII isoform, an N-terminal deletion present in almost 50% of GBMs with receptor amplification [2]. This mutation has been implicated in several aspects of the aggressiveness of gliomas, like the increase in survival/proliferation signals, or the formation of a suitable microenvironment for tumorigenesis, as well as the resistance to therapeutics [31]. The expression of EGFRvIII has been associated with the activation of the downstream signals in the absence of ligands. In addition, it has been proposed that there is an abnormal receptor trafficking, which contributes to increased signaling [8]. However, the data presented here indicates that the activation of AKT activation and the orthotopic tumor growth of GBM cells carrying the EGFRvIII mutation are also sensitive to *TMEM167A* downregulation. It is important to emphasize the high metabolic dependence of cells expressing this oncogene, as they need to maintain high levels of autophagy to tolerate the metabolic stress [32]. A greater autophagy flow allows the recycling of toxic products and the generation of more ATP to keep the proliferative rate. Our results suggest that *TMEM167A* downregulation might alter the coordination between EGFR–AKT signaling and autophagy, which could be crucial for EGFR-dependent GBMs, even in the presence of the vIII activating mutation. Future experiments are warranted in order to characterize the relevance of vesicular acidification in EGFR signaling in the presence of other point mutations.

Overall, our results highlight the relevance of the endomembrane system regulation by TMEM167A in EGFR-dependent gliomas, because it controls a key step in the development of aggressive glioma, where EGFR signaling and autophagy maintenance converge. It would be interesting to test whether other endocytosis-related receptors, expressed in different glioma subtypes, could also be modulated by the levels of TMEM167A in a p53-dependent manner. Moreover, it would be worth testing if molecules like BFA or other v-ATPase inhibitors could have a therapeutic value in gliomas, especially in wild-type p53 tumors. 

## 4. Materials and Methods 

### 4.1. Human Samples 

Glioma tissues samples (Appendix A) were obtained from surgeries at Hospital 12 de Octubre (Madrid, Spain), after patient’s written consent and with the approval of the Ethical Committee (Comité de Etica de la Investigación (CEI) del Hospital 12 de Octubre) (CEI 14/023). Mutations and copy number variations in *TP53*, *PTEN*, and *EGFR* were identified by using a next-generation sequencing (NGS) panel (Ion Torrent technology, ThermoFisher Scientific, Waltham, MA, USA) [33].

### 4.2. Human Glioma Cells 

U87 (EGFRwt, p53wt, PTEN null) and U373 (EGFRwt, p53mut, PTEN null) cells were obtained from the ATCC. GBM2 (EGFRamp, p53mut, and PTENwt) cells were kindly donated by Rosella Galli (San Raffaele Scientific Institute, Milano, Italy). The rest of the human cells (GBM3 (EGFRwt, p53mut, and PTENwt) and GBM4 (EGFRamp/vIII, p53wt, and PTENwt) were obtained by dissociation of surgical specimens from patients treated at Hospital 12 de Octubre (Madrid, Spain). We digested fresh tissue samples enzymatically using Accumax (Merck Millipore, Burlington, MA, USA), and the cells were grown in Complete Media (CM): Neurobasal supplemented with B27 (1:50) and GlutaMAX (1:100) (ThermoFisher Scientific, Waltham, MA, USA); penicillin-streptomycin (1:100) (Lonza Group AG, Basel, Switzerland); 0.4% heparin (Sigma-Aldrich, St. Louis, MO, USA); and 40 ng/mL of EGF and 20 ng/mL of bFGF2 (Peprotech, Rocky Hill, NJ, USA). 

### 4.3. DNA Constructs and Lentiviral/Retroviral Production 

The lentiviral vectors pTRIPZ (shControl) and pTRIPZ-shTMEM167A (shTMEM167A a and b) were used to produce conditionally interfered cells. Infected cells were selected with 1 μg/mL of puromycin. Then, shRNA expression was induced by 1 μg/mL of Dox (Sigma-Aldrich, St. Louis, MO, USA) in vitro or by adding 2 mg/mL of Dox to the drinking water of the mice. The lentiviral vector to express the mutant of p53 was pLenti6/V5-p53_R273H (#Plasmid 22934, Addgene, Watertown, MA, USA). Infected cells were selected with 1 μg/mL of Blasticidin. TMEM167A-GFP vectors were used to transfected 293T cells. 

To obtain the virus, the 293T cells were transiently co-transfected with 5 µg of appropriate lentivector plasmid, 5 µg of packaging plasmid pCMVdR8.74 (#Plasmid 22036, Addgene), and 2 µg of VERSUSV-G envelope protein plasmid pMD2G (#Plasmid 12259, Addgene), using Lipofectamine Plus reagent (Invitrogen, Carisbad, CA, USA). Lentiviral supernatants were prepared by transfection of 293T cells and collection of the culturing media after 48 h. 

### 4.4. Mouse Xenografts

Animal experiments were reviewed and approved by the Research Ethics and Animal Welfare Committee at our institution (Instituto de Salud Carlos III, Madrid, Spain) (PROEX 244/14 and 02/16), in agreement with the European Union and national directives.

Orthotopic xenografts: Stereotactically guided intracranial injections in athymic nude Foxn1nu mice (Harlan Interfauna Iberica, Barcelona, Spain) were performed by administering 1 × 10^5^ cells resuspended in 2 μL of culture medium. The injections were made into the striatum (coordinates: A-P, −0.5 mm; M-L, +2 mm; D-V, −3 mm; related to Bregma), using a Hamilton syringe, and the animals were sacrificed at the onset of symptoms. 

Heterotopic xenografts: Cells (3 × 10^6^) were resuspended 1:10 in culture media and Matrigel (Becton Dickinson, Franklin Lakes, NJ, USA) and then subcutaneously injected into athymic nude Foxn1 nu mice. The tumor volume was measured with a caliper every 7 days. Tumor volume was calculated as ½ (length × width)^2^.

Mice had 2 mg/mL of Dox in their drinking water to induce shRNA expression 2 weeks after the cell injection. Animals were sacrificed by cervical dislocation, and the tumors induced were removed and either fixed in 4% PFA for 24 h before IF staining or freshly frozen for RNA or protein extraction. 

### 4.5. EGFR Signaling Assay

Cells were maintained in serum-free medium overnight. The next day, cells were stimulated with 100 ng/mL of EGF for different indicated times. Then, cells were collected and lysed for WB analysis. 

### 4.6. Flow Cytometry 

Tumor cells (U87, U373, GBM2, GBM3, and GBM4) were disaggregated with Accumax (15 min, room temperature), and then they were stained with an antibody against EGFR conjugated with FITC (Abcam, Cambridge, UK, #ab11400) diluted in PBS-1% BSA (Staining buffer) for 30 min on ice. Cells were washed in PBS, treated with propidium iodide (5 μg/mL, Sigma-Aldrich) and analyzed by flow cytometry (FACSCalibur, Bio-Rad Laboratories, Hercules, CA, USA), using the FlowJo software (https://www.flowjo.com). 

### 4.7. Western Blot Analysis

For protein expression analysis, cultured cells, human samples, or mouse tumor tissue were processed by mechanical disruption in lysis buffer (Tris–HCl pH 7.6, 1 mM EDTA, 1 mM EGTA, 1% SDS, and 1% Triton X-100), followed by heating for 15 min at 96 °C. Protein content was quantified by using a BCA Protein Assay Kit (Thermo Fisher Scientific). Approximately 30 µg of proteins were resolved by 10% or 12% SDS-PAGE, and they were then transferred to a nitrocellulose membrane (Hybond-ECL, Amersham Biosciences, Little Chalfont, UK). The membranes were blocked for 1 h at room temperature in TBS-T (10 mM Tris–HCl (pH 7.5), 100 mM NaCl, and 0.1% Tween-20) with 5% skimmed milk, and then incubated overnight at 4 °C, with the corresponding primary antibody (rabbit anti-Cathepsin B 1:1000, Santa Cruz Biotechnology, Dallas, TX, USA #sc-13989), mouse anti-GAPDH (1:1,500, Santa Cruz Biotechnology #sc-47724), rabbit anti-LC3II (1:2500, Sigma #L8919), rabbit anti-pSer473-AKT (1:1,000, Cell Signaling Technology, Danvers, MA, USA #4060), rabbit anti-AKT (1:1,000, Cell Signaling #4691), rabbit anti-pTyr1068-EGFR (1:1,000, Cell Signaling #3777), mouse anti-p62 (1:1000, Becton Dickinson #610832), and mouse anti-p53 (1:500, Santa Cruz Biotechnology #sc-126) diluted in TBS-T. After being washed 3 times with TBS-T, the membranes were incubated for 2 h at room temperature with their corresponding secondary antibody (HRP-conjugated anti-mouse (#NA931) or anti-rabbit (#NA934), Amersham Biosciences) diluted in TBS-T. Proteins were visible by enhanced chemiluminescence with ECL (Thermo Fisher Scientific), using the Amersham Imager 680 (Amersham Biosciences). Full scans of the WBs are shown in Appendix A.

### 4.8. QRT-PCR Assay 

We extracted RNA from culture cells, human samples or mouse tumor tissue by using an RNA isolation Kit (Roche, Basel, Switzerland). Total RNA (1 µg) was reverse transcribed with a Prime Script RT Reagent Kit (Takara Bio Inc, Kusatsu, Japan). Quantitative real-time PCR was performed by using the Light Cycler 1.5 (Roche) with the SYBR Premix Ex Taq (Takara). The primers used for each reaction were *TMEM167A*: Fw-AGTATGCTGTATAGTAATGG; Rv-ATTCAATGTTCGGAGATAA and *HPRT*: Fw-TGACACTGG CAAAACAATGCA; Rv-GGTCCTTTTCACCAGCAAGCT. We analyzed gene expression data by the ΔΔCt method.

### 4.9. Immunofluorescent Staining 

The 293T cells were grown in DMEM 10% FBS over coverslips and then fixed in 4% paraformaldehyde for 20 min. Cells were blocked for 1 h in 1% FBS and 0.1% Triton X-100 in PBS and incubated overnight with the primary antibody, mouse anti-PI (3,4) P2 (1:100, # Z-P034B, Echelon Biosciences Inc (Salt Lake city, UT, USA). Anti-rabbit IgG Cy3 (1:200, Jackson Immunoresearch, West Grove, PA, USA) secondary antibodies were used, and DNA was stained with DAPI.

For the Lysotracker assay, U87 and U373 cells were grown in DMEN 10% FBS over coverslips and incubated with LysoTracker Red or LysoTracker Blue (1:500, Invitrogen) for 15 min, washed in PBS, and immediately imaging was done with Leica SP-5 confocal microscope. 

### 4.10. Gene Expression and Survival Analyses in Silico

The expression of different genes and the follow-up overall survival data from human glioma tumors corresponding to TCGA or Rembrandt datasets were downloaded from Xena cancer Browser (https://xenabrowser.net) and Gliovis (http://gliovis.bioinfo.cnio.es), respectively. Kaplan–Meier survival curves were done following patient stratification, using gene expression values. The significance of the differences in overall survival between groups was calculated by using the log-rank test as Mantel–Cox (chi-square). The distribution of genetic alterations was analyzed with the TCGA datasets, using cBioportal (www.cbioportal.org). 

### 4.11. Statistical Analysis

All analyses were performed with the GraphPad Prism 5 software. The *p*-values < 0.05 were considered significant (**p* < 0.05; ***p* < 0.01; *** *p* < 0.001; **** *p* < 0.0001; n.s. = nonsignificant). Error bars represent standard error of the mean. For bar graphs, the level of significance was determined by a two-tailed unpaired Student’s *t*-test. The difference between experimental groups was assessed by Paired *t*-Test and one-way ANOVA. For Kaplan–Meier survival curves, the level of significance was determined by the two-tailed log-rank test.

## 5. Conclusions

We have confirmed that there is a mutually exclusive distribution between mutations in *EGFR* and *TP53* in gliomas. Contrary to what occurs in other cancers, our results indicate that wild-type p53 gliomas are more aggressive than their mutant counterparts. Moreover, there is a stronger activation of EGFR signaling in p53 wild-type gliomas, induced by amplifications and/or mutations of the receptor’s gene and by the modulation of EGFR trafficking by TMEM167A. We have found that the expression of TMEM167A, a vesicle-associated protein, has a positive prognostic value in p53 wild-type glioma patients. Downregulation of *TMEM167A* impaired the proper acidification of the endo-lysosomal system and inhibited the activation of AKT in response to EGFR activation, as well as the proper function of the autophagy system. This effect was mimicked by the vacuolar ATPase inhibitor, bafilomycin A1, and was reversed in the presence of p53 mutations. On the basis of these data, we propose that the increased aggressiveness of wild-type p53 gliomas might be due to the increase in growth factor signaling activity, which depends on the regulation of vesicular trafficking by TMEM167A.

## Figures and Tables

**Figure 1 cancers-12-00208-f001:**
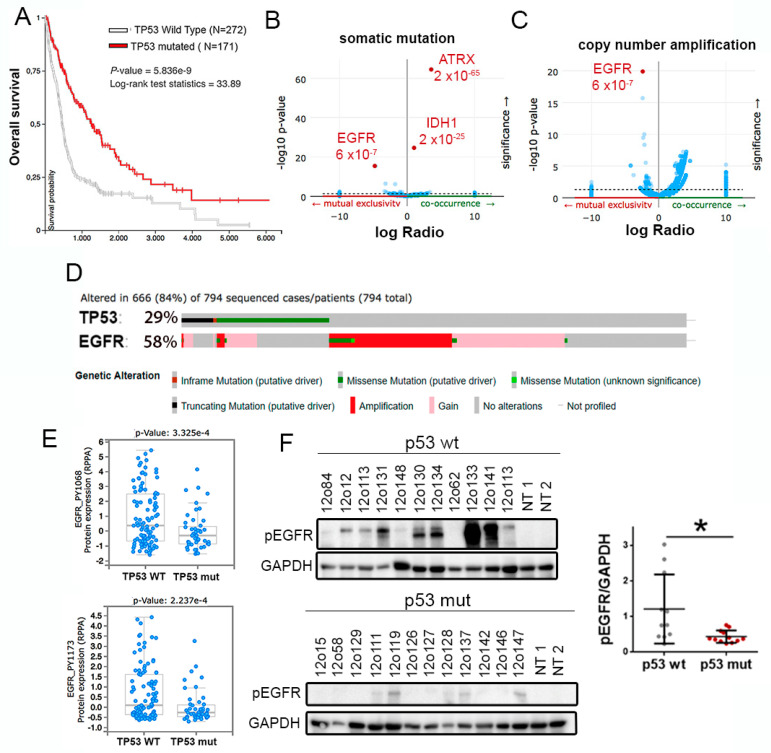
Opposite distribution of TP53 and EGFR alterations in gliomas. (**A**) Kaplan–Meier overall survival curves of patients from the TCGA LGG + GBM cohort. Patients were separated on the basis of the status of TP53. (**B**,**C**) Volcano plots showing mutated genes. (**B**) or copy number amplified genes (**C**) with differential distribution in gliomas, comparing TP53 wild-type and TP53 mutated tumors. (**D**) Distribution of somatic non-silent mutations in EGFR and TP53 in a glioma cohort (TCGA LGG + GBM dataset). (**E**) Analysis of levels of Phospho-Tyr1068-EGFR and Phospho-Tyr1173-EGFR in a cohort of patients with glioma (TCGA LGG + GBM dataset), according to the status of TP53. (**F**) Western blot (WB) analysis and quantification of Phospho-Tyr1068-EGFR (pEGFR) in tumor tissue extracts from patients diagnosed with glioma. Patients were separated based on the status of TP53. GAPDH protein levels were used as a loading control. NT: Normal Tissue. * *p* ≤ 0.05.

**Figure 2 cancers-12-00208-f002:**
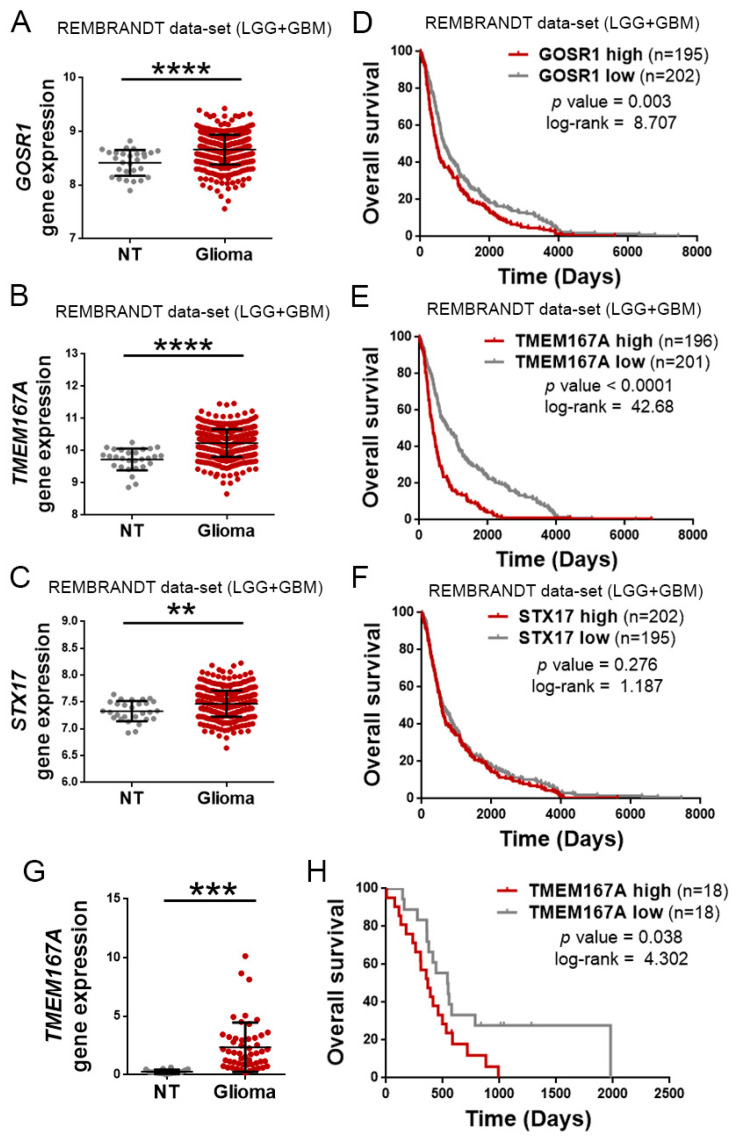
TMEM167A is overexpressed in gliomas and correlates inversely with tumor aggressiveness. Analysis of the amount of GOSR1 (**A**), TMEM167A (**B**), and STX17 (**C**) mRNAs levels in gliomas (LGG + GBM Rembrandt cohort) compared with normal tissue. (**D**–**F**) Kaplan–Meier overall survival curves of patients from the Rembrandt LGG + GBM cohort. Patients were separated on the basis of high and low GOSR1 (**D**), TMEM167A (**E**), and STX17 (**F**) gene expression values. (**G**) qRT-PCR analysis of TMEM167A in our own cohort of glioma samples (*n* = 52) compared with normal tissue. HPRT expression was used for normalization (**H**) Kaplan–Meier overall survival curves of patients from our own glioma cohort (*n* = 36). Patients in each cohort were stratified into 2 groups on the basis of high and low TMEM167A expression values. ** *p* ≤ 0.01, *** *p* ≤ 0.001, **** *p* ≤ 0.0001.

**Figure 3 cancers-12-00208-f003:**
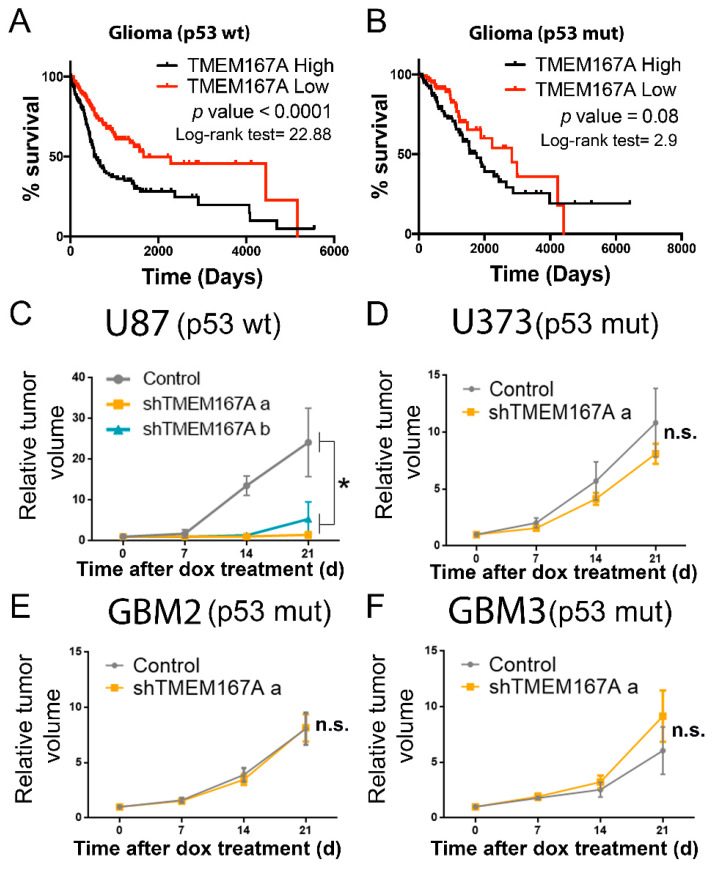
TMEM167A expression correlates with tumor aggressiveness in wild-type but not in mutant p53 gliomas. (**A**,**B**) Kaplan–Meier overall survival curves of patients from the Rembrandt LGG–GBM cohort. Patients were separated on the basis of the TP53 status: p53 wt (**A**) and p53 mut (**B**). Then they were stratified into two groups on the basis of TMEM167A expression values. (**C**–**F**) Subcutaneous tumor growth assay of U87 (**C**), U373 (**D**), GBM2 (**E**), and GBM3 (**F**) cells expressing shControl (Control) or shTMEM167A (shTMEM167Aa and shTMEM167Ab) (*n* = 6/group). The graphs were represented as fold increase in tumor volume. * *p* ≤ 0.05, n.s. = nonsignificant.

**Figure 4 cancers-12-00208-f004:**
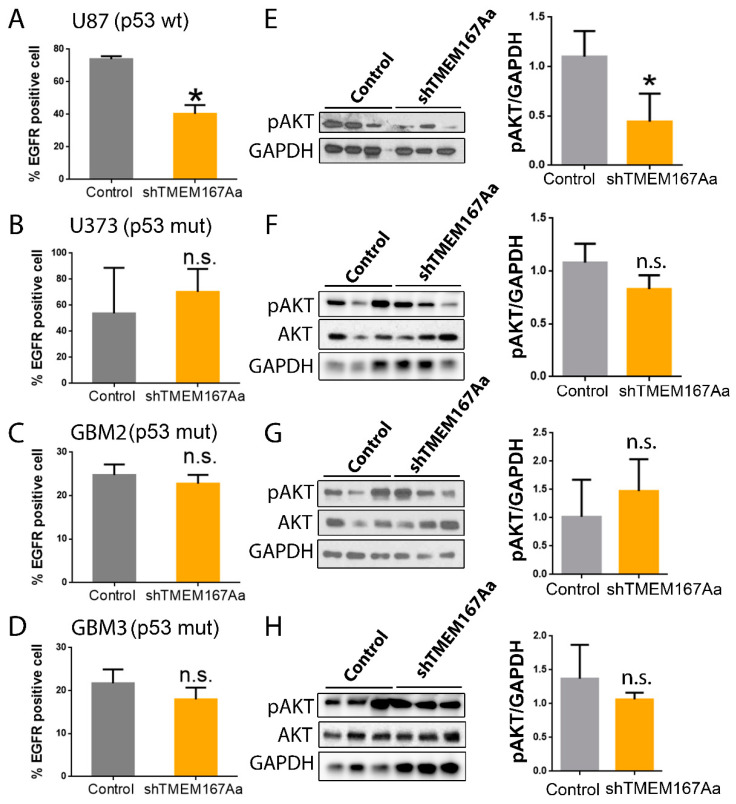
TMEM167A knockdown impairs EGFR stability and AKT signaling in wild-type p53 gliomas. (**A**–**D**) Flow cytometry analysis of the number of EGFR positive cells (%) in subcutaneous tumors from Figure 4C–F. (**E**–**H**) WB analysis and quantification of pSer473-AKT (pAKT) in tumors from Figure 4C–F. GADPH was used as a loading control. * *p* ≤ 0.05, n.s. = nonsignificant.

**Figure 5 cancers-12-00208-f005:**
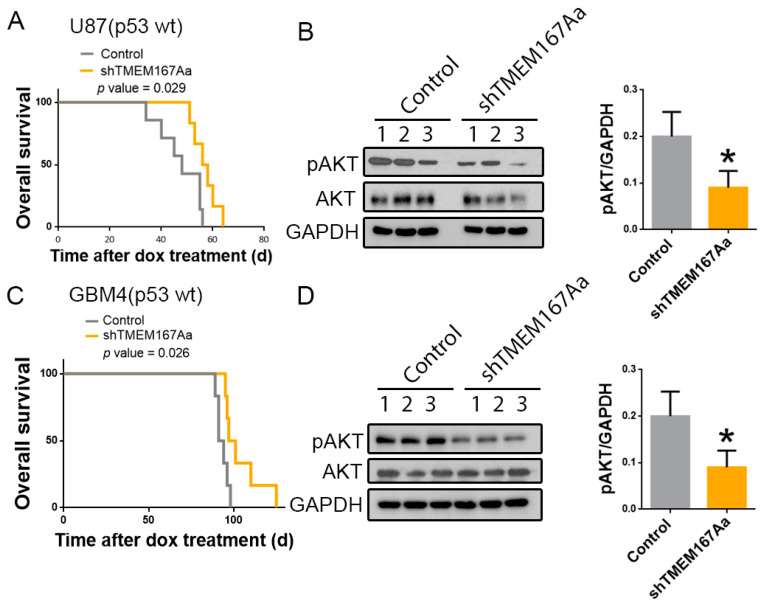
Downregulation of TMEM167A in EGFRwt and EGFRamp/mut cells impairs orthotopic glioma growth and receptor signaling. (**A**–**D**) U87 (**A**,**B**) and GBM4 (**C**,**D**) cells overexpressing shControl (Control) or shTMEM167Aa were injected in the brains of immunodeficient mice (*n* = 6/group). (**A**,**C**) Kaplan–Meier overall survival curves. (**B**,**D**) WB analysis and quantification of pSer473-AKT (pAKT) in three different tumor extracts from A and C, respectively. GADPH was used as a loading control. * *p* ≤ 0.05.

**Figure 6 cancers-12-00208-f006:**
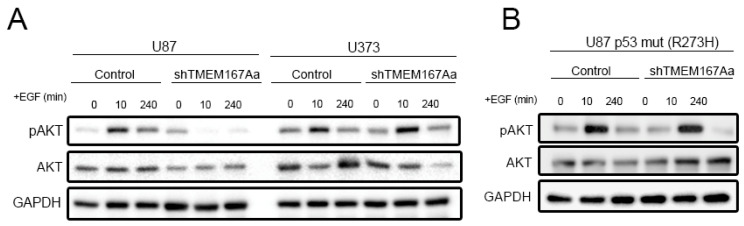
Overexpression of mutant p53 rescues the effect of shTMEM167A. (**A**,**B**) Growth factor-starved U87 and U373 cells (**A**) and U87 Tp53 mut (R273H) cells; (**B**) expressing shControl (Control) or shTMEM167Aa was stimulated with 100 ng/mL of EGF for the indicated times. The amount of pSer473-AKT (pAKT) was tested by WB. GAPDH was used as loading control.

**Figure 7 cancers-12-00208-f007:**
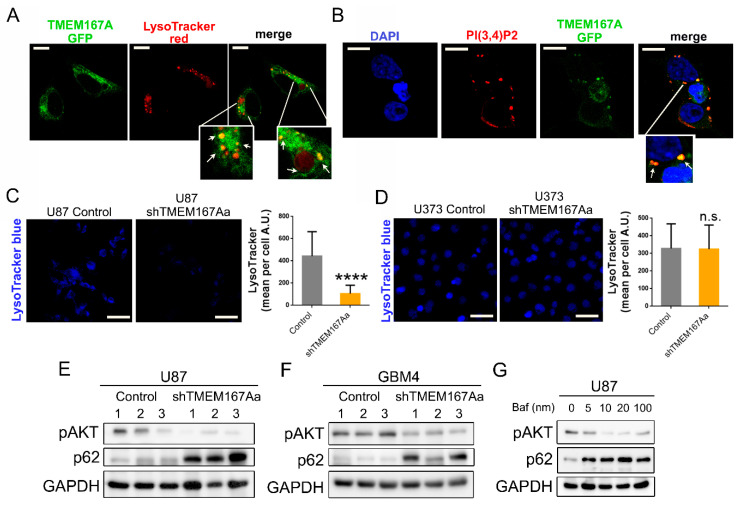
TMEM167A is required for vesicular acidification in wild-type p53 gliomas. (**A**) Representative images of GFP-TMEM167A and LysoTracker co-staining in 293T transfected cells. Arrows point toward colocalizing vesicles. (**B**) Representative images of GFP-TMEM167A and PI (3,4)P2 immunofluorescence (IF) co-staining in 293T transfected cells. Arrows point toward colocalizing vesicles. (**C**,**D**) Confocal images and quantification of LysoTracker staining in U87 (Control or shTMEM167Aa) cells (**C**) and U373 (Control or shTMEM167Aa) cells (**D**). (**E**,**F**) WB analysis of pSer473-AKT (pAKT) and p62 in tumors from Figure 3A (**E**) and Figure 3C (**F**). GADPH was used as a loading control. (**G**) WB analysis of pAKT and p62 in U87 cells treated with Bafilomycin A1 (Baf) at different concentrations. Scale bar: 10 μm (**A**,**B**), 75 μm (**C**,**D**) **** *p* < 0.0001, n.s. = nonsignificant.

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
