# Peer review of "The EGFR-TMEM167A-p53 Axis Defines the Aggressiveness of Gliomas"

_cancers, 2020, doi:10.3390/cancers12010208_

Round 1
Reviewer 1 Report
In this manuscript, the authors examined expression of TMEM167A in gliomas and its roles of cancer progression. Knockdown of TMEM167A inhibits activation of EGF receptor signaling and tumor growth in wild-type p53 gliomas but not in mutant p53 gliomas. Additionally, knockdown of TMEM167A reduces the number of acidic vesicles in wild-type p53 glioma cells. Overall, the experiments were well performed, and the results are interesting. My comments are as follows:
1) My major concern is that the authors used only one cell line (U87) of p53 wild-type glioma cells in this study. Therefore, the authors should use another glioma cell line with wild-type p53 to examine whether knockdown of TMEM167A induces similar effects to those observed in U87 cells.
2) Is TMEM167A expressed at similar levels in p53 wild-type and mutant glioma cells?
3) The authors assess EGF receptor signaling by phosphorylation levels of Akt. Phosphorylation of Akt in gliomas is limited to EGF stimulation, and is not induced by other stimuli?
Author Response
1) My major concern is that the authors used only one cell line (U87) of p53 wild-type glioma cells in this study. Therefore, the authors should use another glioma cell line with wild-type p53 to examine whether knockdown of TMEM167A induces similar effects to those observed in U87 cells.
We are sorry if we have not been clear enough about this issue. We have indeed tested the effect of TMEM167A knockdown in the intracranial growth of two different p53 wild-type glioma cell lines: U87 (old Figure 3A) and GBM4 (old Figure 3C). GBM4 is a primary cell line derived from a human tumor that carries the amplification of EGFR and contains the vIII mutation of this receptor. In this new version of the manuscript we have reorganized the text in order to clarify this point. For that we have changed the order of the Results Sections, explaining first the results in the subcutaneous tumors (with one p53wt cell line (U87) (new Figure 3C) and three p53mut cell lines (U373, GBM2 and GBM3) (new Figures 3D-F), and then the effect of shTMEM167A in the intracranial growth of two p53wt cell lines, U87 (new Figure 5A) and GBM4 (new Figure 5C). To render it clearer we now show the genetic status of p53 and EGFR in these two graphs (Figure 5A, C) and we have included the genetic information of all the cell lines in the Materials and Methods Section. Primary wild-type p53 glioma cells do not grow very well in vitro (based on our own experience and the results from other groups (EMBO J. 2010 Aug 4; 29(15): 2659-74) and they hardly grow in vivo. For that reason we were not able to perform the experiment with GBM4 cells in the subcutaneous setting. We would like to point out that the manuscript also shows the impairment of EGFR/AKT signaling activity after TMEM167A knockdown in the two p53 wild-type gliomas (U87 and GBM4) (new Figure 5B, D and Figure 7E, F).
2) Is TMEM167A expressed at similar levels in p53 wild-type and mutant glioma cells?
Following the reviewer suggestion, we have performed the RT-PCR analysis of TMEM167A in GBM samples (Figure S4A) and xenografts (Figure S4B), grouped based on the presence of wild-type or mutant p53. We have observed that there is indeed less expression in the mutant gliomas, which further supports that the pro-oncogenic function of TMEM167A is no longer needed in mutant tumors. The results are included in Supplementary Figure S3 and cited in the Results Section (line 161).
3) The authors assess EGF receptor signaling by phosphorylation levels of Akt. Phosphorylation of Akt in gliomas is limited to EGF stimulation, and is not induced by other stimuli?
As the reviewer mentions, we have used AKT phosphorylation (P-AKT) as a readout of EGFR signaling. We have previously shown that P-AKT if the most consistent readout of EGFR activity in glioma cells, as it is the most strongly inhibited in the presence of dacomitinib, a specific inhibitor of EGFR kinase activity (Mol Cancer Ther. 2015;14(7):1548-58). As the reviewer suggests, AKT can be phosphorylated in response to other stimuli, especially after the activation of other receptors with Tyrosine-kinase activity (RTKs) or downstream of G-protein coupled receptors (Cell. 2017;169(3):381-405). Here we have shown that TMEM167A controls EGFR, which is one of the most overexpressed genes in gliomas (Cell Mol Life Sci. 2014;71(18):3465-88). Moreover, we had previously observed that the growth of most of the lines used in this study is EGFR-dependent (J Clin Invest. 2013;123(6):2475-87; Mol Cancer Ther. 2015;14(7):1548-58). However, we agree that we cannot discard that TMEM167A might be necessary for the trafficking and/or the signaling of other receptors present in gliomas, as it was already briefly mentioned in the last paragraph of the Discussion Section. We have now included another paragraph referring to this observation (line 311 to 316).
Reviewer 2 Report
Sánchez-Gómez et al. described that the increased aggressiveness of wild-type p53 gliomas might be due to the increase in AKT signaling activity, and it depends on the regulation of vesicular trafficking by TMEM167A.
This is interesting to get an insight of the relationship between p53 status and TMEM167A.
However, it requires major revision as below.
1. In Fig.3B and 3D as well as Fig.5(F,G,H) and Fig.6 total AKT data should be contained as an internal control of western blots. Please check whether GAPDH or AKT in Fig.5 bar graph.
2. In Fig.4E and F, data for GBM with p53 wt must be contained as like GBM with p53 mut.
3. It is also important of PTEN status in GBM as well as p53. Did you check it? Please discuss this point.
4. In materials, catalogue number should be described (e.g. antibodies).
Author Response
Please see attachment.
Reviewer 2
Sánchez-Gómez et al. described that the increased aggressiveness of wild-type p53 gliomas might be due to the increase in AKT signaling activity, and it depends on the regulation of vesicular trafficking by TMEM167A. This is interesting to get an insight of the relationship between p53 status and TMEM167A. However, it requires major revision as below.
1) In Fig.3B and 3D as well as Fig.5(F,G,H) and Fig.6 total AKT data should be contained as an internal control of western blots. Please check whether GAPDH or AKT in Fig.5 bar graph.
We have now included Total AKT and GAPDH as internal controls of the WB analysis. We can confirm that the P-AKT data in the graphs is referred to the amount of GAPDH.
2) In Fig.4E and F, data for GBM with p53 wt must be contained as like GBM with p53 mut.
We are sorry if we have not been clear enough about this issue. We have indeed tested the effect of TMEM167A knockdown in the intracranial growth of two different p53 wild-type glioma cell lines: U87 (old Figure 3A) and GBM4 (old Figure 3C). GBM4 is a primary cell line derived from a human tumor that carries the amplification of EGFR and contains the vIII mutation of this receptor. In this new version of the manuscript we have reorganized the text in order to clarify this point. For that we have changed the order of the Results Sections, explaining first the results in the subcutaneous tumors (with one p53wt cell line (U87) (new Figure 3C) and three p53mut cell lines (U373, GBM2 and GBM3) (new Figures 3D-F), and then the effect of shTMEM167A in the intracranial growth of two p53wt cell lines, U87 (new Figure 5A) and GBM4 (new Figure 5C). To render it clearer we now show the genetic status of p53 and EGFR in these two graphs (Figure 5A, C) and we have included the genetic information of all the cell lines in the Materials and Methods Section. Primary wild-type p53 glioma cells do not grow very well in vitro (based on our own experience and the results from other groups (EMBO J. 2010 Aug 4; 29(15): 2659-74) and they hardly grow in vivo. For that reason we were not able to perform the experiment with GBM4 cells in the subcutaneous setting. We would like to point out that the manuscript also shows the impairment of EGFR/AKT signaling activity after TMEM167A knockdown in the two p53 wild-type gliomas (U87 and GBM4) (new Figure 5B, D and Figure 7E, F).
3) It is also important of PTEN status in GBM as well as p53. Did you check it? Please discuss this point.
We agree with the reviewer that the function of PTEN is very relevant during GBM development (J Neurooncol. 2002 Jun;58(2):107-14.; Cancer Biol Ther. 2008 Sep;7(9):1321-5.; Semin Cancer Biol. 2019 Dec;59:66-79.) and, in particular, for the production of PI(3,4)P2 (Mol Cell. 2017 Nov 2;68(3):566-580.e10.). In this manuscript, we have used PTEN null cell lines (U87, U373) as well as PTEN wild-type gliomas (GBM2, GBM3 and GBM4) (Mol Cancer Ther. 2015 Jul;14(7):1548-58.). This genetic information is now presented in the Materials and Methods Section. In any case, we have not observed any relation between the PTEN genetic status and the sensitivity to TMEM167A downregulation. As an example, we show bellow the lack of expression of PTEN in U87 (sensitive to shTMEM167A) and U373 (non-responsive to shTMEM167A) (see the figure below(in attachment)). We have included a sentence related to this subject in the Discussion section (lines 319 to 325).
4) In materials, catalogue number should be described (e.g. antibodies).
Following the Reviewer suggestion, we have now included the catalog numbers in the Materials and Methods section.

Reviewer 3 Report
The authors here propone a role of EGFR-TMEM167A-p53 axis in defining aggressiveness of gliomas. By analyzing TCGA datasets, they highlight that wild-type p53 gliomas are more aggressive than their mutant counterparts, probably as the result of mutations in EGFR and its consequently stronger activation. They also identify TMEM167A as a gene associated with vesicular trafficking of EGFR in p53 wild-type gliomas and implicated in overall survival in this group of tumors. They provide evidence that TMEM167A knockdown reduce the acidification of intracellular vesicles, affecting the autophagy process and impairing EGFR trafficking and signaling. They conclude that the increased aggressiveness of wild-type p53 gliomas might be due to the increase in growth factor signaling activity, based on the TMEM167A mediated regulation of the vesicular trafficking.
In my opinion, this is a very interesting and well conducted study.
Major concerns:
The majority of data supporting authors’ conclusions were obtained from statistical analysis of specific TCGA datasets, also confirmed by supporting experiments in cell lines and animal models. However, authors also used an in-house cohort of 52 glioma samples. Surprisingly, I didn't find details about investigation of these cases (e.g. how and if they had been characterized from the molecular point of view). This point should be clarified to the reader and more information should also be reported in the materials and methods.
Minor concerns:
I’d like to suggest you a check of some grammar mistake within the text.
I noted the following:
Page 2 (line 71): “downstream target’s” should be ““downstream targets”
Page 2 (line 92): “and in silico” should be “an in silico”
Page 13 (line 426): human simples” should be “human samples”
Author Response
The majority of data supporting authors’ conclusions were obtained from statistical analysis of specific TCGA datasets, also confirmed by supporting experiments in cell lines and animal models. However, authors also used an in-house cohort of 52 glioma samples. Surprisingly, I didn't find details about investigation of these cases (e.g. how and if they had been characterized from the molecular point of view). This point should be clarified to the reader and more information should also be reported in the materials and methods.
Following the Reviewer´s suggestion we have now included the basic genetic information of the human samples used in this study in Table S1. We have also indicated in the Materials and Methods sections the procedures used to characterize molecularly those samples.
Minor concerns:
I’d like to suggest you a check of some grammar mistake within the text.
I noted the following:
Page 2 (line 71): “downstream target’s” should be ““downstream targets”
Page 2 (line 92): “and in silico” should be “an in silico”
Page 13 (line 426): human simples” should be “human samples”
In this new version of the manuscript we have corrected these mistakes
Round 2
Reviewer 1 Report
Authors have adequately addressed my concerns, and I would like to recommend the publication of the manuscript in Cancers.
Reviewer 2 Report
This revised form is now acceptable for the publication of "Cancers".